# Dietary practices, physical activity and social determinants of non-communicable diseases in Nepal: A systemic analysis

Sudesh Sharma[1,2]*, Anna Matheson[3], Danielle Lambrick[4], James Faulkner[5], David W. Lounsbury[6], Abhinav Vaidya[7], Rachel Page[2]

1 DIYASU Community Development Centre, Biratnagar, Morang, Nepal, 2 Massey University, Wellington, Wellington Region, New Zealand, 3 Victoria University of Wellington, Wellington, Wellington Region, New Zealand, 4 University of Southampton, Southampton, Hampshire, United Kingdom, 5 University of Winchester, Winchester, Hampshire, United Kingdom, 6 Albert Einstein College of Medicine, Bronx, New York, United States of America, 7 Kathmandu Medical College, Kathmandu, Bagmati, Nepal

* yoursudesh@gmail.com

**Data Availability Statement:** Transcripts (without any personal identifier and of those who consented to sharing of the interview transcripts) are available

## Abstract

Unhealthy dietary habits and physical inactivity are major risk factors of non-communicable diseases (NCDs) globally. The objective of this paper was to describe the role of dietary practices and physical activity in the interaction of the social determinants of NCDs in Nepal, a developing economy. The study was a qualitative study design involving two districts in Nepal, whereby data was collected via key informant interviews (n = 63) and focus group discussions (n = 12). Thematic analysis of the qualitative data was performed, and a causal loop diagram was built to illustrate the dynamic interactions of the social determinants of NCDs based on the themes. The study also involved sense-making sessions with policy level and local stakeholders. Four key interacting themes emerged from the study describing current dietary and physical activity practices, influence of junk food, role of health system and socio-economic factors as root causes. While the current dietary and physical activity-related practices within communities were unhealthy, the broader determinants such as socio-economic circumstances and gender further fuelled such practices. The health system has potential to play a more effective role in the prevention of the behavioural and social determinants of NCDs.

## Background

Unhealthy dietary habits and physical inactivity have been identified as key risk factors in increasing non-communicable diseases (NCDs) in developing countries [1, 2]. It has been estimated that dietary and physical activity risks attribute to approximately 5.3 million premature deaths annually [3]. In Nepal, the last Stepwise approach to surveillance (STEPS) survey has shown that about 99% of people did not consume sufficient fruits and vegetables but 97% had adequate physical activity [4]. However, the problem of physical inactivity was more pervasive in the urban and peri-urban areas with more than 5% of urban respondents categorised as

on request (Email: yoursudesh@gmail.com; r.a. page@massey.ac.nz; humanethics@massey.ac.nz) after signing confidentiality agreement due to ethical reasons.

**Funding:** SRS studied towards his PhD at Massey University supported by Massey University Doctoral Scholarship and Massey University Graduate Research Support. This paper is part of the doctoral study. The funding support has no role in the study design, data collection and analysis and preparation and publication decisions relating to the study. There is no any other external funding to report.

**Competing interests:** The authors declare that no competing interests exist.

physically inactive [4]. A recent study showed that inadequate physical activity among adolescents was 85% [5].

Increasingly, there is evidence relating to socio-economic and environmental determinants influencing diet and physical activity, often collectively termed social determinants [6, 7]. Global health agencies and experts have been pushing the agenda of NCDs prevention in both developed and developing nations through focusing on the social determinants of health [8–12]. In 2008, the World Health Organization proposed the landmark *Social Determinants of Health Framework*, which highlighted the association of proximal and distal determinants of health and suggested directions for actions [11]. Since then, there has been increasing focus in understanding and taking action on the social determinants of complex health issues such as NCDs globally [13]. However, developing countries are far behind in their understanding and actions on the social determinants of health compared to developed countries [13–17]. On a positive note, some Latin American and Caribbean countries have initiated significant steps in addressing the social determinants of NCDs [18, 19].

Global evidence has suggested that physical inactivity as well as junk food consumption is rapidly increasing due to social, environmental and economic influences [6, 7, 20]. Rapid urbanisation and changing nature of work (sedentary nature and increasing mechanisation of physical work) are impacting physical activity and dietary patterns [6, 21]. A Lancet review has suggested that physical inactivity is increasing among lower socio-economic groups [21]. Moodie et. al. suggested that transnational companies are driving the junk food epidemic globally, resulting in increasing NCDs, even in developing countries [20]. Studies from Nepal have shown that urban population have an increased risk of chronic diseases due to ready access to junk food and limited avenues for physical activity [4, 22–24]. Further, limited policy and multi-sectoral action on healthy diet and optimal physical activity promotion have resulted in increase in NCDs and their risks [2, 25]. However, there is limited evidence or analysis of the links between social determinants and NCDs mediated by diet and physical activity in Nepalese and developing countries context. This paper assesses the role of dietary practices and physical activity in the interaction of the social determinants of NCDs in Nepal.

## Methods

The study design was qualitative study based on systemic intervention (SI) methodology (Fig 1) [26, 27]. SI adopts the critical systems thinking approach and stresses on using multiple methods to understand the complex problems. A combination of case study and system dynamics (causal loop diagram) methods, along with participatory sense-making workshops, were adopted within the research design.

### Study area and study participants

The study was conducted in Nepal from July until October 2016. There was limited district specific data on the prevalence of NCDs, however the nation surveys showed an increasing trend in both urban and rural areas throughout Nepal [4, 28]. Two districts (Bhaktapur and Morang) were purposively selected as case districts, based on first author's prior relationships with respective district health offices and place. In particular, the first author is from Morang and his motivation to study about social determinants of NCDs grew from his personal experience of observing family and wider community being impacted by NCDs. Bhaktapur is a hilly district near the capital city of Nepal and has a sub-tropical cold and humid climate. Morang is a plain district located in the eastern part of Nepal and has a sub-tropical hot and humid climate. Within each case district, one municipality and two village development committees (VDC) were selected for community level data collection.

**Fig 1. Systemic intervention study design to study social determinants of NCDs in Nepal (image republished from authors' open access article [27]).**

Community members affected by NCDs and key stakeholders from community, district and policy level were the participants of the study. The key stakeholders were purposively selected based on their current engagement (or potential engagement) in NCDs prevention and health promotion in the case districts and at policy level in Nepal. They included members of *NCDs Multi-sectoral Action Plan Steering Committee*, academia, non-government organisations, District Health Officer, Local Development Officer, community-based organisations, local health workers and village leaders. The communities for focus group discussion (FGD) were identified through the referral of respective district health offices and local health institutions.

## Data collection

Data collection involved semi-structured interviews with key stakeholders from the case districts and policy level. Policy level data were collected to supplement the information from case districts. Semi-structured interview guidelines so developed were guided by the adapted social determinants of health framework (Fig 2).

The sampling strategy for the study entailed identifying and participating individuals knowledgeable and experienced in the issue of interest i.e. NCDs prevention at case district and policy level in Nepal. The purpose of the purposive sampling was to capture the different perspectives and contexts on the NCDs prevention issue guided by the study framework (Fig 2). Purposive sampling is often used in qualitative studies for identifying and interviewing potential information-rich subjects on a complex issue like NCDs [29]. *Maximum variation* strategy was adopted to ensure diverse perspectives across the health and social sectors in Nepal were captured relating to complex issue of NCDs [29]. Thirty-nine participants were interviewed from the two case study sites (Bhaktapur and Morang) and 24 participants from

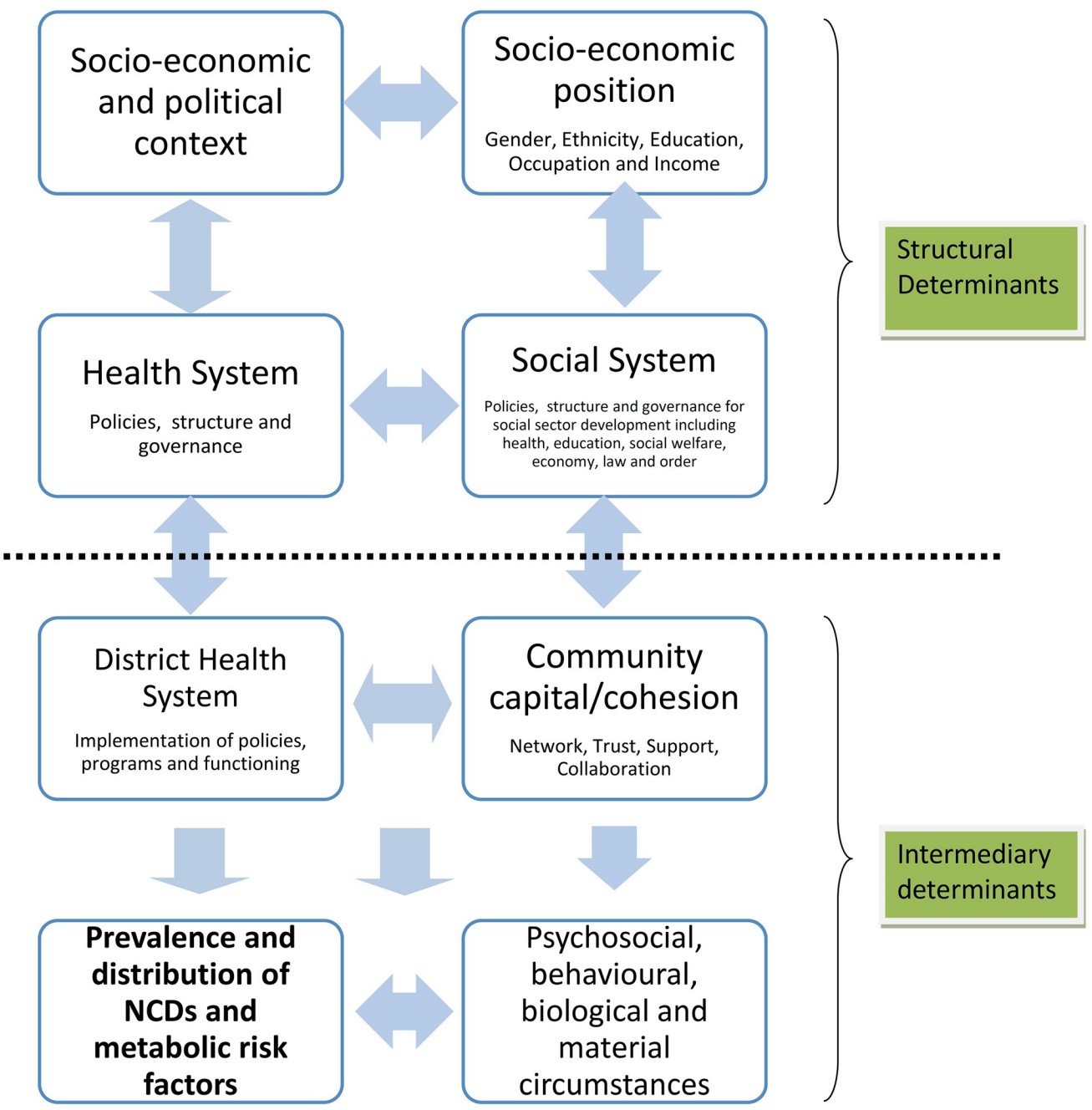

**Fig 2. Adapted Social determinants of health framework for the study of the social determinants of NCDs in Nepal (image republished from authors' open access article [27]).**

policy level. All of the interviews were carried out by the first author in Nepali language. The time of interview ranged from 30 minutes to one hour.

Further, 12 focused group discussions (FGDs) were conducted within the selected municipalities and VDCs of the case districts. Each FGD included five to 10 community people experiencing and/or caring for family members with NCDs and their metabolic risks. The purpose of FGDs was to capture negotiated views on NCDs and risks as experienced by

individuals, families and community members belonging to different socio-economic groups. Therefore, in each VDC/municipality, two FGDs were conducted with one group representing a socio-economically disadvantaged community and the other a more advantaged/mixed community. The FGDs were facilitated by the first author with the help of local volunteers. The duration of FGDs ranged from 45 minutes to one hour.

## Data management and analysis

The interviews and FGDs audio recordings were first transcribed in Nepali and then translated into English for coding and thematic analysis. A *Framework Approach* was used to undertake the thematic analysis guided by the study framework (Fig 2) [30]. Themes were identified, grouped and coded by using both a deductive approach based on the study framework as well as more inductive approaches to allow for emergent themes. Case analysis helped in framing the qualitative and quantitative data for the local context and iterating on association of various social determinants. A systems map or causal Loop Diagram (CLD) was developed to illustrate the association of the social determinants of NCDs indicated by thematic analysis. Causal loop diagramming is a system dynamics technique to build a system map of a problem of interest that illustrates the causal association through feedback structures that are loopy in nature [31]. CLD consists of two types of feedback loops: balancing and reinforcing loops, and these loops have variables connected by uni-directional arrows with either positive or negative signs on them. A positive sign indicates that if an independent variable increases or decreases, the dependent variable also increases or decreases in the same direction. A negative sign indicates the exact opposite meaning i.e. if an independent variable increases, then the dependent variable decreases and vice versa. The balancing loop is a goal-seeking loop, which is indicated by *B* within CLD and indicates the stabilising feature of the loop. In general, causal loops with an intervention are of a balancing nature that tries to mitigate the problem. On the other hand, reinforcing loops (indicated by *R* within CLD) depict a mechanism that produces a result that influences more of the same action, thus resulting in either growth or decline. An example of a reinforcing loop can be the vicious cycle of poverty and ill health. *Dedoose (Socio-cultural Research Consultants)* and *MS Excel 2016 (Microsoft)* [32] were used to manage the qualitative data [33] and *Vensim (Ventana Systems Inc.)* was used to build CLD [34].

## Stakeholder validation

Stakeholder validation was conducted through organising three sense-making workshops, two at respective case districts and one at national level during January/February, 2018. The workshops helped to further enrich the analysis through the feedback and suggestions from the stakeholders. These workshops, indeed, became opportunities to sensitise the stakeholders about the impending epidemic in Nepal. The workshops proved useful in reaching the purpose of this research and the methodological stance of systemic intervention methodology.

## Ethical consideration

Formal ethical approvals were taken from the Massey University Human Ethics Committee (SOA 16/37) and Nepal Health Research Council Ethics Committee (Reg. no. 163/2016) respectively. All participants, and in particular disadvantaged and less literate groups, were clearly informed about the purpose of the study. All participants were provided with an information sheet, and a voluntary written consent was obtained from all participants of semi-structured interviews and focus group discussions.

## Findings

Four key interacting thematic areas relating to dietary practices and physical activity emerged, which were utilised to develop a causal loop diagram and demonstrate dynamic complexity.

### Theme 1: Practice relating to diet and physical activity contributing to increased risk

Key informants and focus group participants indicated that current food habits among most Nepalese were not balanced, which was potentially leading to the problem of NCDs. A focus group participant from rural Morang shared:

*"Nepalese food is taken three times a day. And there is rice every time, even in the snack. That is why now, 2 out of 4 people of a family have high sugar level."*

*(ID: 69)*

Some participants expressed pre-existing and predominant beliefs, which was sustaining such unhealthy dietary practices.

*"We eat enough food (at night). Since that has to withstand us up to 10 am in the morning."*

*(ID: 69; Rural Morang; FGD Participant)*

One key observation highlighted by participants from the case districts was that the traditional practice of healthy eating was being gradually displaced by westernised practice of eating, which is unhealthy for the Nepalese context. A FGD participant from rural Bhaktapur lamented:

*"We used to have "Dhido"[thick porridge made from corn or buckwheat flour] in the morning and in the evening. We used to run home from school during lunchtime, have the lunch and return back to school. But these days vehicles come to pick up and drop our children and there is no time to prepare lunch for the children. So junk foods are easy."*

*(ID: 76)*

A policy level key informant explained that overwhelming availability and appeal of junk foods were affecting local dietary practices.

*"In case of food habit, our raw food/fruits habits have been changed to junk food based diet. Junk food are readily available, easy to cook and do not need much effort. There is an increase in the use of junk food from a very early age."*

*(ID: 3; Policy Stakeholder; Health)*

Some participants stressed that the Nepalese culture and traditions have potential to promote physical activity and healthy diet but they continue to be ignored. Offering prayers to different temples in a single walk (Newari culture), yoga and meditation (Hindu culture), *Namaz* prayer (ritual Muslim prayer, which also involves different body movements) were stressed by participants as practices that have physical activity significance.

*"In context of our geographically and culturally diverse country, we have our own traditions of practicing yoga, meditations, bipasyana etc. but these aren't practiced in our daily schedule."*

*(ID: 12; Policy Stakeholder; Health)*

Participants also reported the decline in physical activities like walking due to increasing use of motor vehicles in the communities.

*"People who used to walk now use bicycle and motorbike now and that makes easy for them. Because of that too, it has been decreased now."*

*(ID: 66; Urban Morang; FGD Participant)*

A case study district stakeholder elaborated that people often resort to physical activity practice, including yoga and morning walks, after they start suffering from diseases or metabolic risk.

*"If we ask 100 people who come to Yoga if they do not have any such disease or conditions, only few hands would raise up."*

*(ID: 30; Bhaktapur District Stakeholder; Non-health)*

## Theme 2: Health sector issues delaying actions to improve dietary practice and physical activity

Key policy stakeholders shared that the health sector has historically considered NCDs as a curative care agenda instead of a multi-sectoral prevention agenda. This has led to the current lack of concrete structure and leadership relating to NCDs prevention which is creating a delay in coordinated action for nutrition and physical activity promotion.

*"Our health system is resisting to adapt to addressing NCDs issues from health promotion and prevention aspect."*

*(ID: 18; Policy Stakeholder; Health)*

As a result, district stakeholders reported that the district health system and below have limited NCDs prevention-related activities in addition to systemic health system issues. Health workers lacked training in NCDs prevention and the office had limited resources at their disposal. A district stakeholder shared how the current health system was more focused on curative care than preventative action.

*"Our [Primary Health Care] network is very good. But, it is less focus on primary prevention compared to curative. Our system can do more primary prevention."*

*(ID; 25; Bhaktapur District Stakeholder; Health)*

A key policy stakeholder expressed limited health sector intervention in curbing marketing and promotion of junk food leading to current junk food based dietary practice and displacing the traditional diet.

*"Second one is food habit where major issue is aggressive advertisement of junk foods. What is happening is, gradually we are leaving our indigenous food habits and shifting*

*towards the trend of consumption of junk foods may be as a fashion or may be as a status symbol."*

*(ID*: 9; Policy Stakeholder; Non-health)

A rural health worker shared how the dietary practice has shifted in the last 10 years as follows:

*"Nowadays people are eating lot of fast food. Ten years earlier people used to have home beaten rice and beans; corn (pop-corn) and would have a balance diet. But now people do not eat at home; they eat outside and mostly junk food."*

*(ID*: 35; Rural Bhaktapur; Health Worker)

In addition, participants stressed that increased motorisation and lack of systemic infrastructure were leading to limited physical activity in the case studies. There has been a rapid development in road infrastructure and motorisation of public and private transport systems. However, due to lack of awareness regarding active travel and luxury-seeking behaviour, this has been contributing to physical inactiveness to some extent.

*"These days, vehicles are available right in our doorsteps. Now instead of walking to the place that takes 30 minutes, we wait an hour for the vehicles to take us there."*

*(ID*: 65; Urban Morang; FGD Participants)

Urban areas of Nepal have poor urban planning, leading to limited green and open spaces for promoting physical activity among the residents. Though there are policies and regulations relating to urban and housing design in place, residents simply ignore them, and the regulations are not strongly implemented enough to have any meaningful impact. This is leading to cumulative effect of limited space as explained by a policy stakeholder:

*"What has happened is lack of open space has become an increasing problem because of population growth and exposure to open space is diminishing /decreasing. This leads to decrease in people's habit of roaming, walking, exercise and lack of good/high oxygen concentration. This trend is slowly fading away even in rural areas and condition has worsened in urban areas."*

*(ID*: 11; Policy Stakeholder; Health)

Some participants linked limited physical activity with system ignorance and limited promotion of yoga and traditional physical activity. Often the system was not focusing on instilling habits of physical activities among the young people.

*"We ourselves are responsible for the current state of yoga. We couldn't direct the young generations to a correct track."*

*(ID*: 52; District Stakeholder Morang; Non-health)

## Theme 3: Gendered-related norms and cultural shifts affecting physical activity and dietary practice

There were some revelations about the influence of gender on physical activity and dietary practice. Participants shared that in urban areas, females tended to be physically inactive and overweight/obese.

*"In urban and semi urban areas obesity is seen more in married female."*

*(ID*: 5; Policy level)

However, there were some positive actions being initiated within urban communities where more females compared to male were engaged in yoga.

*"You may see that many females come to the yoga and are often married. Not many males come; probably due to complacency and laziness."*

*(ID*: 44; Urban Bhaktapur; Social Worker)

Similarly, participants shared that more and more families of foreign employed males were migrating to urban centres for a better life. As a result, female members within such families were detached from their regular agricultural works in the rural context and confined to limited space rented houses, which significantly reduced their physical activity.

*"If we talk about those families whose members are working abroad, their families now have started living in the market areas and depend upon market for every necessities including the vegetables which once they themselves used to grow."*

*(ID*: 7; Policy Stakeholder; Health)

In rural areas of case districts, women who used to be engaged in agricultural activities and animal rearing have rapidly given up such practices.

*"In our time we used to run to the field soon after having the meal. We used to cultivate different crops and plants like maize, mustard and many more. We used to grow mustard here like in Chitwan but now there are only bushes and grasses in those lands."*

*(ID*: 76; Rural Bhaktapur; Female FGD Participant)

## Theme 4: Migration affecting agriculture sector and exposing vulnerable families to obesogenic environment

The participants pointed out that migratory movement of youths and adults and their families from rural to urban and foreign countries was one of the key reasons for the rapid decline of agriculture activities. The migration created a shortage of agricultural workforce in rural areas. The migration was mainly driven by limited economic opportunities in rural areas and prospect of better opportunities in urban and foreign land. A policy stakeholder noted that foreign migration was having double whammy effects in terms of affecting the agriculture sector as well as exposing vulnerable families to unhealthy lifestyles.

*"At least one member of a household is earning in foreign land. Now the proportion of population involved in agriculture has decreased in recent times. People's profession has changed. Along with the change in profession, people's lifestyle has also changed."*

*(Policy Stakeholder; ID*: 6; Health)

A rural stakeholder further added that the decline in agricultural activities was directly resulting in increased dependency on commercially available food and decrease in physical activity in rural areas.

*"Importantly, agricultural activity is on a decline, everyone is now buying and eating food items in rural areas. People do not engage in hard agricultural practices as they used to do. People now eat too much of unhealthy products."*

*(ID: 55; Rural Morang; Health Facility Management Committee member)*

One of the key stakeholders explained that despite unhealthy dietary practice and physical inactivity being widespread, vulnerable groups were disproportionately affected in terms of health effects.

*"There are certain things about poor population as they have unhealthy diet but they are involved in lots of physical activity during digging, working; the disease differs between the poor and the rich. Physical activity is the good aspect among the poor but there are many other factors that are negative like unhealthy diet (consume more fast foods), inaccessibility of treatment facility and many stress factors among the poor people results in their diseased condition. Beside that behaviour of smoking and alcohol consumption among poor is another cause of disease."*

*(Policy Stakeholder; ID: 4; Health)*

### CLD of major sub-systems depicting the interaction of the social determinants of unhealthy diet and physical inactivity and NCDs

The CLDs below have been derived from the thematic analysis and show the dynamic interaction of the key variables discussed above contributing to the problem of NCDs in Nepal. As indicated, the development of the CLDs was an iterative process and was conducted simultaneously with thematic analysis. The CLDs depict three prominent feedback mechanisms or sub-systems displaying the interactions of the social determinants of NCDs: prevention delay, demand-supply and socio-economic influence.

In Fig 3, the prevention delay sub-system consists of delayed balancing loops, which highlight themes that indicate limited and delayed action by the health system towards addressing unhealthy diet and physical inactivity-related practices as indicated in the results. Awareness-raising campaigns and multi-sectoral coordination for promoting sustainable food systems and physical activity-friendly environments were lacking for prevention of NCDs. The health system was possibly guided by the mental model that unhealthy diet and physical inactivity were more due to individual behaviour, hence the interventions in developing countries were limited to campaigns and medical services provision. The CLD highlighted the themes indicating that multi-sectoral coordination for sustainable food systems and improving physical infrastructure were continually neglected by the overall health and social system. The themes also indicated that junk food companies are using various tactics to promote their products as a healthier option. The same sub-system hints at the curative orientation of the health system (i.e. the increasing chronic diseases problem in turn puts pressure on the government to provide curative care services, leaving limited resources for preventative actions, including multi-sectoral coordination).

Demand-supply sub-system (Fig 4) highlights how individuals and communities were being framed for consuming junk food and drinks due to limited policy and programmatic interventions relating to junk food/drink. The demand-supply sub-system mainly comprised of reinforcing loops and showed that junk food availability was resulting in increased

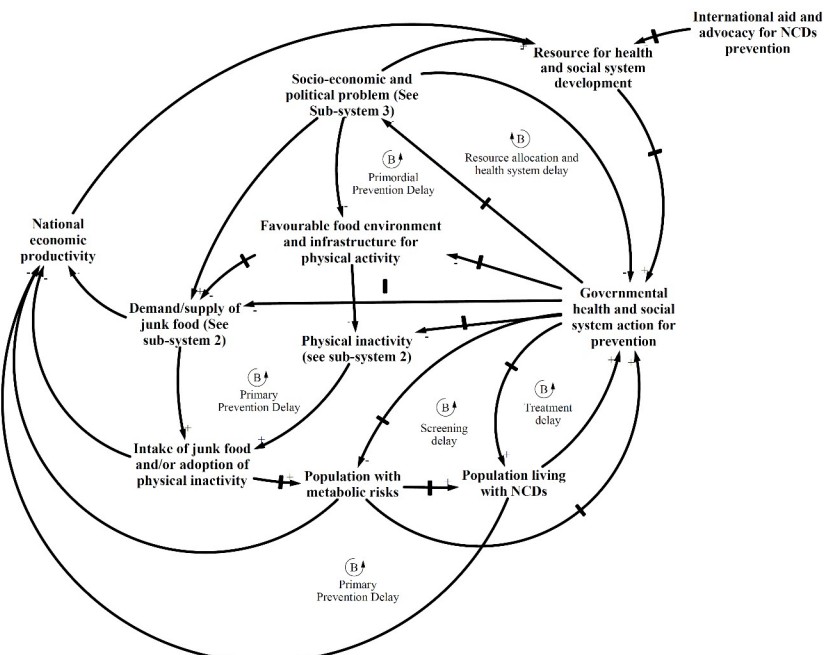

**Fig 3. Prevention delay sub-system showing the interaction of governmental response to promote healthy diet and physical activity, other sub-systems and NCDs.**

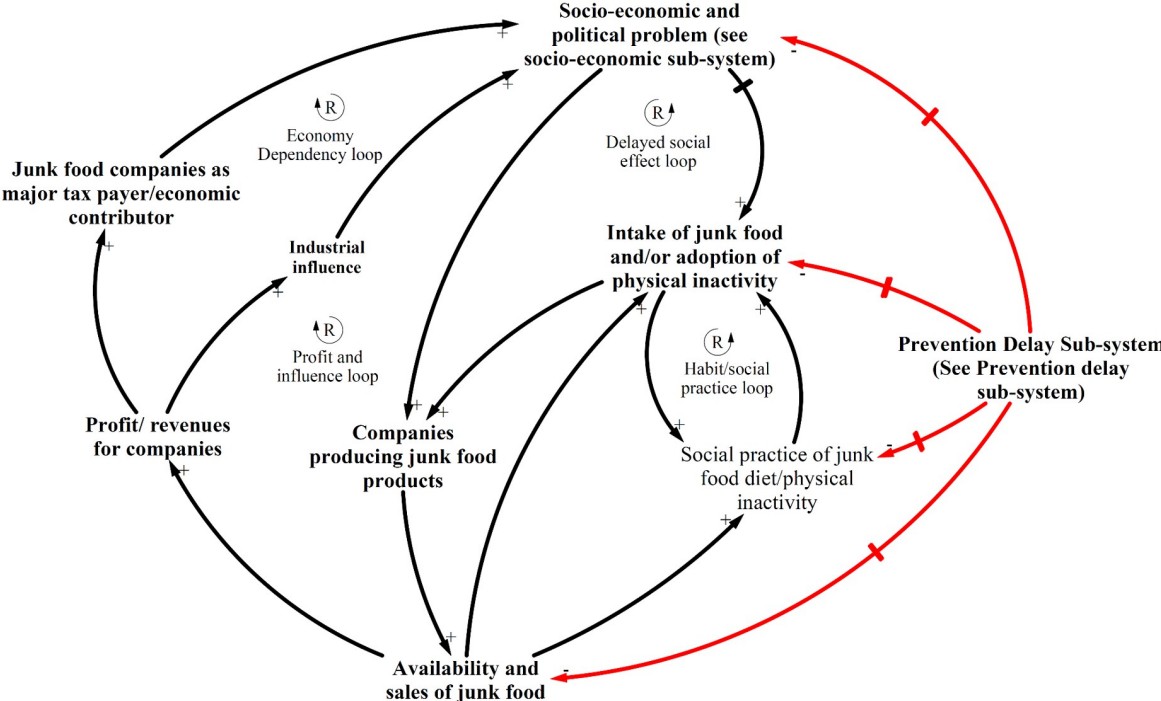

**Fig 4. Demand-supply sub-system showing the interaction of availability and sales of junk food, other sub-systems and NCDs.**

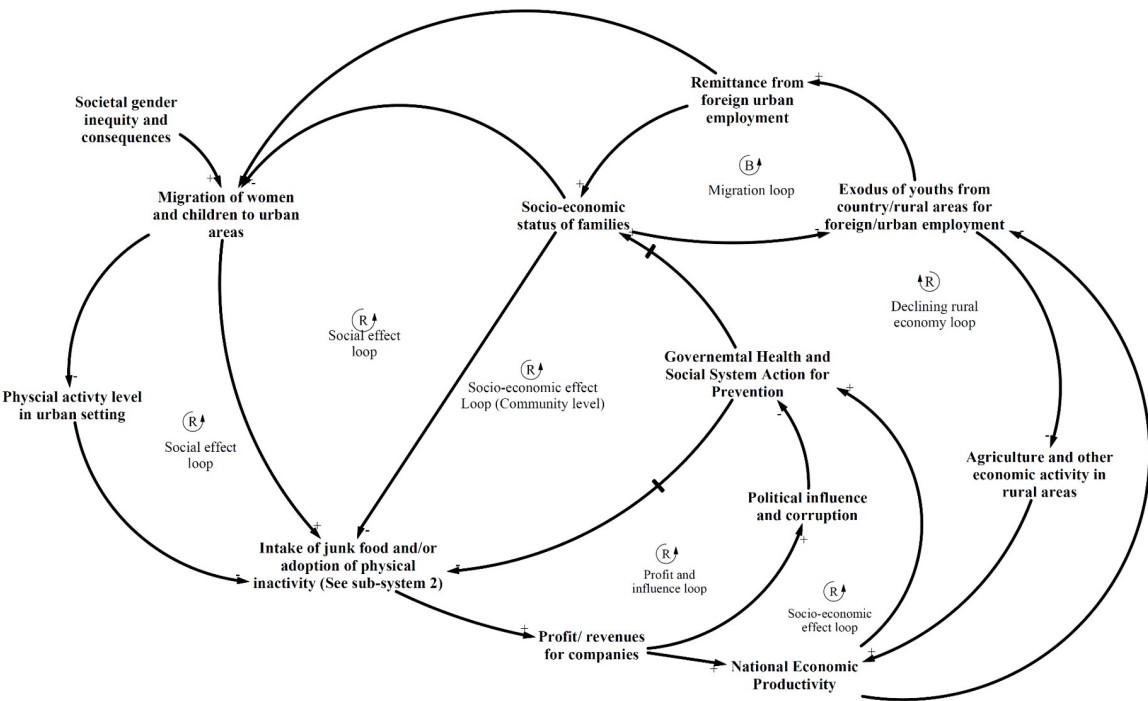

**Fig 5. Socio-economic influence sub-system showing the interaction of socio-economic status, other sub-systems for dietary and physical activity practices and NCDs.**

consumption and affecting local food systems as well as social practices in the case district of Nepal. The extensive marketing of such products as healthy, easy to prepare, cost-effective, family food, etc. have driven the consumption of junk food. The junk food companies are possibly utilising their economic strength to influence both government and public through policy influences and marketing as indicated by study participants.

Socio-economic sub-system (Fig 5) can be conceptualised to distally drive the other two sub-systems towards increasing the burden of NCDs as per the thematic analysis. The sub-system shows the dynamic complexity about how junk foods, physical inactivity and social and economic influences were interacting, which was giving rise to NCDs problems. In particular, socio-economic situation was fuelling the migration of vulnerable groups from rural to city areas and foreign countries, which was not only pushing the rural agriculture system to the verge of collapse but also exposing rural people to unhealthy lifestyles. The sub-system also shows the disproportionate vulnerability and health effects experienced by the female members of migrant families and females in general.

## Discussion

The causal association of the diet and physical activity with NCDs was complex and is discussed here in the light of existing evidence and from a systems perspective. Specifically, we utilise system archetypes (Fig 6), a simpler version of CLDs, to present the mechanism in a way that generates insights for health system action [35].

### Promotion of healthy dietary behaviour and physical activity

In this study, the prevention delay sub-system indicated that the health system has not been structured and functioning effectively to promote healthy behaviours relating to diet and

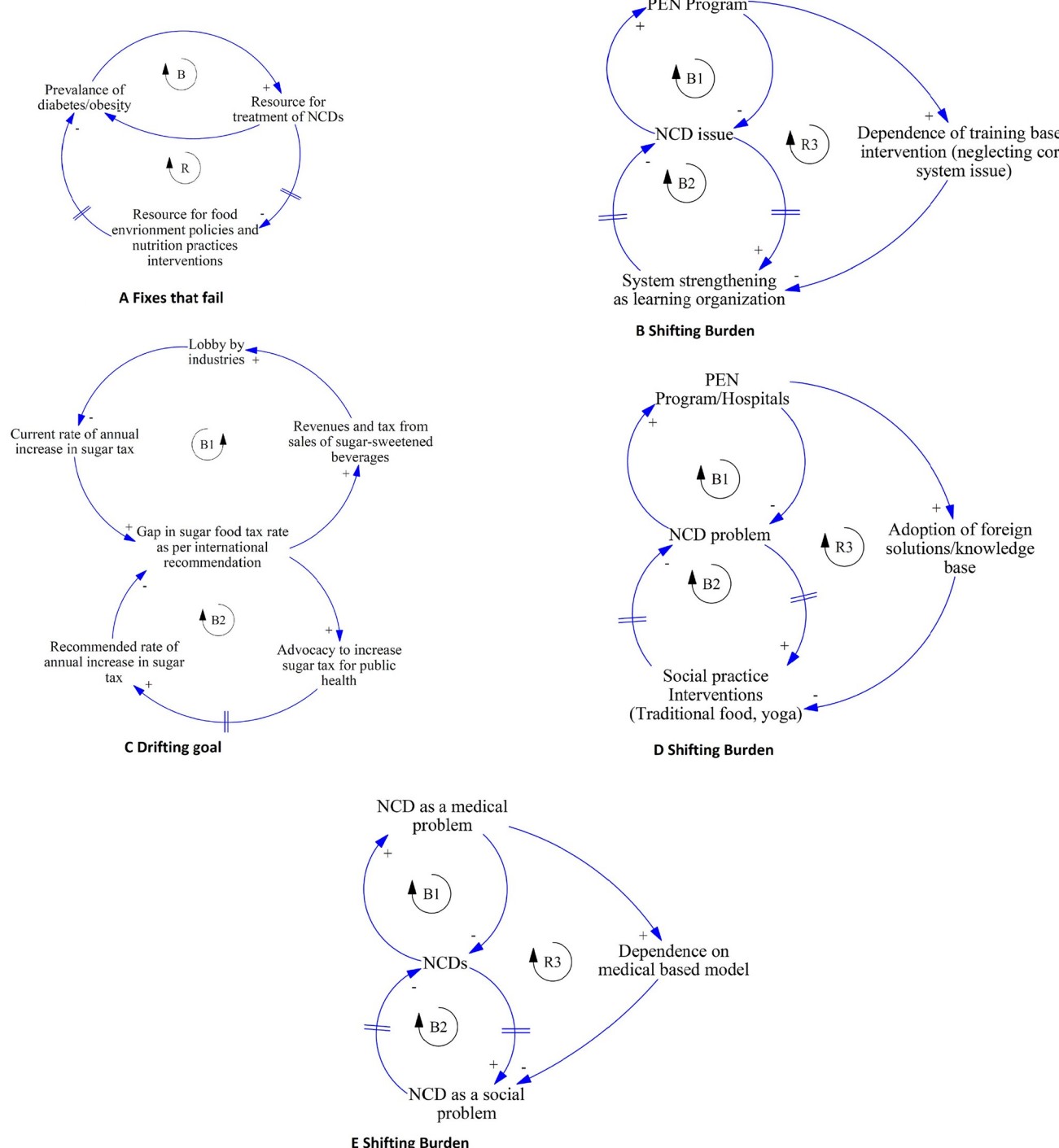

**Fig 6. System archetypes showing different mechanisms of influence of dietary and physical activity practices on NCDs.**

physical activity. The health system in many developing countries like Nepal is yet to prioritise the prevention of NCDs and their behavioural risks [36–39]. In Nepal, evidence indicates that NCDs have been considered as a curative service agenda with increasing budgetary provisions in hospitals, patient care and development of clinical human resources rather than prevention [40]. This resonates with the *Fixes that Fail* system archetype (Fig 6A).

Further, in many developing countries, national strategies for improving diet and physical activity exist [41–44] but most strategies are not effectively implemented due to limited resources and system inefficiency [45, 46]. In particular, developing countries are often lagging behind in terms of formulating and implementing legislations and programmes relating to control of junk food as well as developing physical activity-friendly infrastructures [20, 47], similar to the findings in this study. While developed countries are gradually addressing the junk food epidemic through policy measures, developing countries do not have any mechanism in place to control the junk food epidemic. The junk food companies are being increasingly compared to tobacco/alcohol companies due to their marketing strategies that target vulnerable children and population groups and their role in affecting the local food systems and ultimately health [20]. Further, poor systematic infrastructure-related policy action has been contributing to inadequate physical activity around the world [47, 48]. Prevention delay sub-system has illustrated that disease prevention and health promotion efforts in Nepal have been offset by systemic issues within the health system. As a result, complex problems like NCDs continue to persist and there is a tendency of the health system to focus on easily implementable solutions rather than a comprehensive approach which requires health system strengthening. This can be captured by the *Shifting Burden* archetype (Fig 6B). Many developing countries face similar health system challenges such as poor governance and management, resulting in inefficiency and ineffectiveness [49–52]. The health system continues to be ineffective and inefficient due to top-down approach, and leadership and management issues. Public health actions in developing countries are often short-sighted and implemented without collaborating effectively with other sectors. Strong leadership, appropriate system structure and political commitment could play a significant role in strengthening the health system and supporting comprehensive prevention initiatives in developing countries like Nepal [46, 53]. This shift will ensure more time and resources for the implementation of the prevention initiatives that are essential for combating complex problems like NCDs and their behavioural risks [54, 55].

### Junk food access and marketing influencing consumption of junk foods

Demand-supply sub-system showed an increased access to highly processed food by everyone, especially young people. Evidence suggests that the junk food habit is very pervasive among children and adults in both urban and rural areas of South Asia region [56–59]. Further, a key mechanism that was prominent in the findings was the deskilling of traditional food preparation due to wider availability of junk food and fast food. Evidence from around the world has also confirmed that traditional-based diets are gradually being replaced by junk food habit [60–63]. A review has highlighted how the local food systems were being shaped more by profit motives of the companies than by optimal nutrition [64]. In addition, migration of male youths, especially from rural areas, was driving the migration of families to urban areas, which is suddenly exposing families to an obesogenic environment.

Marketing of junk food is another key reason for the increasing access of junk food as shown by different types of evidence [65] and highlighted by the demand-supply sub-system. Often, the junk food companies through quality advertisements would associate their products with some social services or claim their products to be nutritious and can be enjoyed by an entire family [66]. Such practices of social marketing and misleading vulnerable populations have been well documented in developed and developing countries [66–68]. Similar to the findings of this study, a report identified the price factor, easy availability and lack of government control were some of the factors that were leading to families providing children with junk food [66].

Enforcement agencies have limited ability to control junk food due to lack of regulations and positive framing of junk food by companies producing junk food. Junk food and sugar-sweetened beverages issues have not received much attention in public health policies and action in developing countries [20], which has also been reflected in the themes of this paper. Further, the effects of junk foods on local food systems have not been explored extensively. Developed countries with ample evidence of the effects of junk food are still facing stiff challenges in preventing the junk food induced obesity epidemic [69, 70]. The labelling of products is still a contested issue in Australia and New Zealand [71]. A Lancet review suggested that tactics of food companies influencing national policies can be considered similar to that of tobacco companies [20]. The *Drifting Goal* system archetype depicts the influence of food industries in delaying public health regulations for junk food such as sugar tax (Fig 6C).

## State of physical activity including traditional exercises

As noted in the themes and indicated in the CLDs (interaction between all three subsystems), physical activity was on a decline in both urban and rural areas. A study in India showed that physical activity was gradually decreasing in rural India as well due to increasing urban influences, including adopting urban lifestyle in rural settings, motor vehicle access and mechanisation of agricultural activities in rural areas [72]. Research in rural India and Bangladesh show a similar increasing phenomenon of physical inactivity in rural areas [73–75]. This study showed a shift in social practices (such as travelling, working and exercise practices) resulting in adults being physically inactive, which was further complicated by the habit of consuming energy-dense food. A similar shift in diet and physical activity has been observed elsewhere in developing countries contributing to NCDs [76–78]. People only resorted to healthy diet and exercise after being affected with metabolic risk, a tendency that can be explained by behavioural and psychological theories [79, 80]. Evidence from South Asia region evidence has even suggested that people do not readily change their lifestyle even after being affected with NCDs metabolic risks [81].

The study findings suggested that traditional exercises were being ignored, which have potential to keep the general population healthy, based on cultural values and beliefs. The Vedic culture of Nepal (relating to Hindu religion) is rich in such exercise practices, particularly in yoga and meditations. Yoga is becoming increasingly popular in the Western countries due to its benefits [82] but continues to be neglected in Nepal. This can be explained by *Shifting the Burden* archetype (Fig 6D) where the health system has not realised the most effective and preventive measures are based on local socio-cultural practices.

## Socio-economic situation, migration and gender effect on dietary practice and physical activity

This study showed that socio-economic factors were influencing the diet and physical activity practices in multiple ways within the case districts. In rural areas, there was limited awareness about diet and physical activity. A relatively higher level of physical activity in rural areas could be attributed to agriculture and farming practices, while in urban areas, despite awareness, practice of balanced diet and physical activity was lacking. Available evidence also indicated this complex relationship [21, 83]. Some evidence showed that higher income groups were physically inactive while in some context, lower income groups had less physical activity [84, 85]. A Lancet article noted this complexity and indicated that there may be a social patterning, which was shifting in the developing economy [21]. Such a shift of social patterning needs to be further researched. In this study, health was secondary to the vulnerable group who were experiencing economic hardship. More and more people are migrating to urban areas due to

economic hardship and unequal development, which is exposing people to urban influences of junk food diet and limited physical activity. A study in the US has shown that low-income households have a fatalistic attitude towards their health, possibly due to socio-economic circumstances and lack of ability to control their lives [86].

This study also found an increasing issue of unhealthy diet and physical inactivity among women, both in urban and rural areas. While evidence to support our findings from rural areas was limited, evidence suggests women in urban areas are more vulnerable to the obesogenic environment than male [58]. Urban females' lack of exercise could be understood by the fact that they remain mostly inactive due to confinement to household duties and child rearing, an outcome of Nepalese patriarchal social construct. A sense of the low power of females can be interpreted by limited access, time and autonomy for females to care for their health and engage in active lifestyles. Similar findings of limited autonomy among women relating to diet and physical activity related decisions have been reported elsewhere [87]. This study, particularly, identified the increased health vulnerability of rural women who migrated to urban areas due to sudden exposure to urban lifestyle. A more critical observation is needed to understand the level of autonomy and access to services of the women among different socio-economic groups in both rural and urban areas.

The study findings indicated that the rapid migration and resulting decline in rural agricultural activity had both health and economic implications. The migration of economically active youths and their families to cities and foreign countries not only affected their lifestyle but also affected the local food systems and economy. A review has suggested that locally grown fresh foods were being replaced by imported vegetables and international junk food and supplies, which are cheaper [88]. As a result, the share of agriculture in the gross domestic product has been rapidly decreasing despite a significant (but declining) population still being engaged in agriculture and availability of improved technologies for the agriculture sector [89].

Further, the interaction of the socio-economic sub-system with the prevention delay sub-system (as shown in CLDs) is yet to be appreciated by the health system, thus resulting in limited action on social determinants of NCDs. This continuous ignorance of socio-economic determinants of NCDs by the health system closely resonates with *Shifting the Burden* archetype (Fig 6E). This CLD model and its archetypes could provide critical insights to the health system in addressing NCDs through comprehensive multi-sectoral action to address those socio-economic determinants of unhealthy diet and physical inactivity. Health sector is in a good position to advocate and accelerate actions on the social determinants of NCDs in relation to diet and physical activity and these CLDs and archetypes could be a starting point for action.

The research comprised of some key limitations. The broader scope of the study framework meant that we could only partially illustrate the key social determinants of unhealthy diet and physical inactivity risks in the context of Nepal. We have indicated within the paper that localised qualitative and quantitative studies are needed to understand the dynamics interactions of the social determinants of NCDs and their key risk factors. Further, the CLDs can be improved as new evidence emerges, which is one of the key strengths of systems thinking approaches to understand and tackle a complex problem. A key methodological challenge was limited participation of local stakeholders during the sense-making workshops, which did affect the quality of sense-making workshops.

## Conclusion

This paper has presented a combination of key social determinants influencing dietary practices and physical activity in Nepal. Health system actions were delayed in terms of regulating junk food sales and coordinating with other sectors for improving physical activity. While diet

and physical activity-related knowledge and practices as well as environmental factors were limited, the broader determinants such as socio-economic status accelerated the migration to urban areas and exposure of population (in particular, female and children) to an obesogenic environment. In particular, traditional dietary and physical activity practices were being displaced negatively affecting one's health. Decline in agricultural activities was affecting physical activity level, local food systems and ultimately economy. The health system could play a key role in raising awareness, monitoring the markets for unethical marketing of junk food, and coordinating with other sectors for strengthening local food systems, creating infrastructure that promotes physical activity and addressing socio-economic determinants of unhealthy diet and physical inactivity. The Ministry of Health could advocate and show evidence of the need for such preventative action through the creation of a powerful national agency to facilitate inter-sectoral collaboration and achieve SDH focus.

## Supporting information

**S1 Checklist. COREC checklist.**
(RTF)

**S1 File. Interview and FGD guidelines.**
(DOCX)

## Acknowledgments

The paper is derived from the PhD project of the first author and hence would like to acknowledge all the participants and supporters of the PhD study including Ministry of Health/Nepal, District Public Health Office Morang, District Public Health Office Bhaktapur. The authors would also like to thank research ethics committees of Massey University and Nepal Health Research Council. We would like to acknowledge Mr. Shiva Raj Mishra (PhD Student at University of Queensland) and Mr. Mohan Paudel (PhD Student at Flinders University) for their critical review and suggestions on this paper.

## Author Contributions

**Conceptualization:** Sudesh Sharma.

**Data curation:** Sudesh Sharma.

**Formal analysis:** Sudesh Sharma, Anna Matheson, Danielle Lambrick, James Faulkner, David W. Lounsbury, Abhinav Vaidya, Rachel Page.

**Methodology:** Anna Matheson, Danielle Lambrick, James Faulkner, David W. Lounsbury, Abhinav Vaidya, Rachel Page.

**Supervision:** Anna Matheson, Danielle Lambrick, James Faulkner, David W. Lounsbury, Abhinav Vaidya, Rachel Page.

**Writing – original draft:** Sudesh Sharma.

**Writing – review & editing:** Sudesh Sharma, Anna Matheson, Danielle Lambrick, James Faulkner, David W. Lounsbury, Abhinav Vaidya, Rachel Page.

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
