## [Decision Letter · Decision Letter 0]

14 Sep 2021

PONE-D-21-14408Dietary practices, physical activity and social determinants of non-communicable diseases in Nepal: a systemic analysisPLOS ONE

Dear Dr. Sharma,

Thank you for submitting your manuscript to PLOS ONE. After careful consideration, we feel that it has merit but does not fully meet PLOS ONE’s publication criteria as it currently stands. Therefore, we invite you to submit a revised version of the manuscript that addresses the points raised during the review process.

We look forward to receiving your revised manuscript.

Kind regards,

Maria G Grammatikopoulou

Academic Editor

PLOS ONE

Journal Requirements:

2. We noted in your submission details that a portion of your manuscript may have been presented or published elsewhere. [This article is different from a related published (https://bmcpublichealth.biomedcentral.com/articles/10.1186/s12889-020-09446-2) article in terms of findings and discussion. Since the article is based on my PhD thesis, articles that have been published (or will be published) have similar methods section (and hence the use of same figure relating to methodology and study framework).] Please clarify whether this [conference proceeding or publication] was peer-reviewed and formally published. If this work was previously peer-reviewed and published, in the cover letter please provide the reason that this work does not constitute dual publication and should be included in the current manuscript.

Reviewers' comments:

Reviewer's Responses to Questions

**Comments to the Author**

1. Is the manuscript technically sound, and do the data support the conclusions?

Reviewer #1: Partly

Reviewer #2: Yes

2. Has the statistical analysis been performed appropriately and rigorously? 

Reviewer #1: Yes

Reviewer #2: N/A

3. Have the authors made all data underlying the findings in their manuscript fully available?

Reviewer #1: No

Reviewer #2: Yes

4. Is the manuscript presented in an intelligible fashion and written in standard English?

Reviewer #1: Yes

Reviewer #2: Yes

5. Review Comments to the Author

Reviewer #1: This is a very interesting and mostly well-written paper. The authors utilize appropriate methods to draw their conclusions

Some minor comments

1. Please consider using sex instead of gender, given you did not record one's gender. In case you measured one's gender, this should be more clearly presented in the methods

2. The manuscript requires some English editing

3. Avoid using terms such as "causal relattionship", "influence" etc. Given your study design cannot support these claims, please replace with association etc

Reviewer #2: Very timely and useful research. Probably, on of the most critical public health issue of our time is the growing burden of non-communicable disease. Most of these diseases are preventable and heavily dependent on behavior change. Changing people’s behavior requires a thorough understanding of the root causes of behaviors and health system-related structural issue and the interlinkages with social-determinants of health leading to NCDs. The authors clearly described and organized their findings along these lines. Just two question.

1. Why were the study areas purposively selected? What was the rational for this judgment sampling rather than random selection? Was it to save time? I understand the application of the method to identify participants within the selected communities, as you stated because of their knowledge and experience. But, why were the communities selected through this method?

2. Most of the findings do not seem new. It would be helpful, if the authors emphasized on what is the key new knowledge versus what was already known. For example, we know urbanization, increasing availability and consumption of junk food, alcohol consumption, smoking and reduced physical activities are risk factors for NCDs. Is the key on the findings the Nepalese context?

6. PLOS authors have the option to publish the peer review history of their article (what does this mean?). If published, this will include your full peer review and any attached files.

Reviewer #1: No

Reviewer #2: No

---

## [Author Response · Author response to Decision Letter 0]

12 May 2022

Response to reviewers

- Thank you for your suggestion. We have reviewed and reformatted the manuscript as per the PLOS ONE style.

2. We noted in your submission details that a portion of your manuscript may have been presented or published elsewhere. [This article is different from a related published (https://bmcpublichealth.biomedcentral.com/articles/10.1186/s12889-020-09446-2) article in terms of findings and discussion. Since the article is based on my PhD thesis, articles that have been published (or will be published) have similar methods section (and hence the use of same figure relating to methodology and study framework).] Please clarify whether this [conference proceeding or publication] was peer-reviewed and formally published. If this work was previously peer-reviewed and published, in the cover letter please provide the reason that this work does not constitute dual publication and should be included in the current manuscript.

- This is the second paper from the PhD thesis and includes results about diet and physical activity. The first paper concentrated on results from the PhD focussing on tobacco and alcohol use (https://bmcpublichealth.biomedcentral.com/articles/10.1186/s12889-020-09446-2) which has been peer-reviewed and published. Since the first published paper and this paper on diet and physical activity are from the first authors PhD thesis, we have used and referenced figures relating to the methods and study framework from the published paper in the submitted paper focusing on diet and physical activity. The introduction, findings and discussion are entirely different to the published paper and have not been published elsewhere and hence, we do not feel this constitutes dual publication. 

. 

- ORCID ID of the corresponding author (SS): https://orcid.org/0000-0002-7880-5517

- The reference list has been reviewed and is complete and correct. 

Reviewers' comments:

Reviewer's Responses to Questions

Comments to the Author

1. Is the manuscript technically sound, and do the data support the conclusions?

Reviewer #1: Partly

Reviewer #2: Yes

Thank you for considering our manuscript as scientifically and technically sound.

2. Has the statistical analysis been performed appropriately and rigorously?

Reviewer #1: Yes

Reviewer #2: N/A

We agree that statistical analysis is not applicable for this study.

3. Have the authors made all data underlying the findings in their manuscript fully available?

Reviewer #1: No

Reviewer #2: Yes

We agree with reviewer 1 that still there is limitation in data availability and have shared that with editors. We are in the process of making data available public once we review the transcripts and make sure all personal identifiers of the respondents are removed, as Nepal is a small country and participants from policy level are easily identified. We hope to make the data publicly available by this year.

4. Is the manuscript presented in an intelligible fashion and written in standard English?

Reviewer #1: Yes

Reviewer #2: Yes

Thank you.

5. Review Comments to the Author

Reviewer #1: This is a very interesting and mostly well-written paper. The authors utilize appropriate methods to draw their conclusions

Some minor comments

1. Please consider using sex instead of gender, given you did not record one's gender. In case you measured one's gender, this should be more clearly presented in the methods

- Thank you for your kind suggestion and we agree that we have not measured gender within this paper. However, gender reflects socially constructed characteristics of male and female. We have indeed provided some context such as Nepal being patriarchal society and having more social and economic power as a breadwinner of the family. Also, our findings and discussion are linking those gendered experiences with diet and physical activity related practices. Thus, we feel that it makes more sense if we keep the term “gender” within the manuscript. 

2. The manuscript requires some English editing

- The English has been reviewed by co-authors who are native English speakers.

3. Avoid using terms such as "causal relationship", "influence" etc. Given your study design cannot support these claims, please replace with association etc

- Thank you for your kind suggestion. We have revised and replaced “causal relationship” with “association” accordingly. However, the term “Causal Loop Diagrams” is a standard method/tool name and we have not changed it. Also, as this is a qualitative study, keeping “influence(s)” made more sense (qualitative term) than replacing with “association” (quantitative term) and hence have decided to keep the term “influence (s)” as it is. 

Reviewer #2: Very timely and useful research. Probably, on of the most critical public health issue of our time is the growing burden of non-communicable disease. Most of these diseases are preventable and heavily dependent on behavior change. Changing people’s behavior requires a thorough understanding of the root causes of behaviors and health system-related structural issue and the interlinkages with social-determinants of health leading to NCDs. The authors clearly described and organized their findings along these lines. Just two question.

1. Why were the study areas purposively selected? What was the rational for this judgment sampling rather than random selection? Was it to save time? I understand the application of the method to identify participants within the selected communities, as you stated because of their knowledge and experience. But, why were the communities selected through this method?

- This is a qualitative study and often, cases are purposively chosen. In the study, the main purpose was to choose different context i.e. urban, semi-urban and rural settings, and was also selected in consultation with District Health Offices. Time and convenience of the first author (principal investigator) also influenced the choice of districts which is acceptable in purposive sampling. In qualitative studies, often random sampling is not the choice of sampling.

2. Most of the findings do not seem new. It would be helpful, if the authors emphasized on what is the key new knowledge versus what was already known. For example, we know urbanization, increasing availability and consumption of junk food, alcohol consumption, smoking and reduced physical activities are risk factors for NCDs. Is the key on the findings the Nepalese context?

- The findings are especially important in Nepalese context where the focus is on changing behaviour. Yes, the findings may not be new for high-income countries but, for low income countries like Nepal, the authors hope that this paper can provide a whole new way of seeing a public health issue, taking a systems perspective and designing appropriate actions. This paper provides qualitative evidence on social determinants of dietary and physical activity practices which is very scarce in Nepalese (or low income country) and we hope that this provides some systemic insights for actions by concerned health agencies in Nepal.

6. PLOS authors have the option to publish the peer review history of their article (what does this mean?). If published, this will include your full peer review and any attached files.

Do you want your identity to be public for this peer review? For information about this choice, including consent withdrawal, please see our Privacy Policy.

Reviewer #1: No

Reviewer #2: No

---

## [Decision Letter · Decision Letter 1]

25 Oct 2022

PONE-D-21-14408R1Dietary practices, physical activity and social determinants of non-communicable diseases in Nepal: a systemic analysisPLOS ONE

Dear Dr. Sharma,

Thank you for submitting your manuscript to PLOS ONE. After careful consideration, we feel that it has merit but does not fully meet PLOS ONE’s publication criteria as it currently stands. Therefore, we invite you to submit a revised version of the manuscript that addresses the points raised during the review process.

We look forward to receiving your revised manuscript.

Kind regards,

Sandra Boatemaa Kushitor, Ph.D.

Academic Editor

PLOS ONE

Journal Requirements:

Additional Editor Comments:

Abstract

Incomplete sentence, add diet and physical activity after determinants

....interactions of the social determinants ....

Can the authors provide how this can be achieved:

The health system has potential to play a more effective role in the prevention of the behavioural and social determinants of NCDs.

I have made a few comments.

Reviewers' comments:

Reviewer's Responses to Questions

**Comments to the Author**

1. If the authors have adequately addressed your comments raised in a previous round of review and you feel that this manuscript is now acceptable for publication, you may indicate that here to bypass the “Comments to the Author” section, enter your conflict of interest statement in the “Confidential to Editor” section, and submit your "Accept" recommendation.

Reviewer #3: (No Response)

2. Is the manuscript technically sound, and do the data support the conclusions?

Reviewer #3: Yes

3. Has the statistical analysis been performed appropriately and rigorously? 

Reviewer #3: N/A

4. Have the authors made all data underlying the findings in their manuscript fully available?

Reviewer #3: Yes

5. Is the manuscript presented in an intelligible fashion and written in standard English?

Reviewer #3: Yes

6. Review Comments to the Author

Reviewer #3: (No Response)

7. PLOS authors have the option to publish the peer review history of their article (what does this mean?). If published, this will include your full peer review and any attached files.

Reviewer #3: **Yes: **Dr. Krishna Paudel

---

## [Author Response · Author response to Decision Letter 1]

9 Dec 2022

We would like to thank reviewers for their helpful suggestions and feedback for improving the manuscript.

We authors have reviewed and checked the references and links and have addressed the comments as follows:

Abstract

Incomplete sentence, add diet and physical activity after determinants

....interactions of the social determinants ....

- We have added “of NCDs” as we are being consistent throughout the article.

Can the authors provide how this can be achieved:

The health system has potential to play a more effective role in the prevention of the behavioural and social determinants of NCDs.

- This statement is from abstract. We authors have elaborated this in the conclusion section. As indicated within the paper, curative orientation and health system management issues as well as political and commercial influences outside of the health system are creating systemic barriers for effective multi-sectoral action for the prevention of NCDs risk factors. The first and most important shift needed is creating and resourcing an appropriate prevention structure (a national centre) for acclerating actions on the SDH. 

I have made a few comments (as HTML markup)

- Thank you for your suggestion. We authors have reviewed and revised the manuscript as per HTML markup as follows:

- Line 7 strikeout: changed percentage and statement so that the contrasting argument is more clearer i.e. “but 97% had adequate physical activity”

- Line 10: deleted as suggested i.e. “indicating the shifts towards physical inactivity in Nepal”

- Line 53-59: As suggested, further clarified rationale for selecting the districts i.e. “The study was conducted in Nepal from July until October 2016. While there was limited district specific data on the prevalance of NCDs, the nation surveys showed an increasing trend in both urban and rural areas throughout Nepal [4, 28]. Two districts (Bhaktapur and Morang) were purposively selected as case districts, based on first author’s prior relationships with respective district health offices and place. In particular, the first author is from Morang and his motivation to study about social determinants of NCDs grew from his personal experience of observing family and wider community being impacted by NCDs.”

- Line 170: Nepali term “Dhido” explained within large bracket i.e. “thick porridge made from corn or buckwheat flour”

---

## [Editor Report · Decision Letter 2]

23 Jan 2023

Dietary practices, physical activity and social determinants of non-communicable diseases in Nepal: a systemic analysis

PONE-D-21-14408R2

Dear Dr. Sharma,

We’re pleased to inform you that your manuscript has been judged scientifically suitable for publication and will be formally accepted for publication once it meets all outstanding technical requirements.

Kind regards,

Sandra Boatemaa Kushitor, Ph.D.

Academic Editor

PLOS ONE
---

## [Editor Report · Acceptance letter]

26 Jan 2023

PONE-D-21-14408R2 

Dietary practices, physical activity and social determinants of non-communicable diseases in Nepal: a systemic analysis 

Dear Dr. Sharma:

I'm pleased to inform you that your manuscript has been deemed suitable for publication in PLOS ONE. Congratulations! Your manuscript is now with our production department. 

Kind regards, 

on behalf of

Dr. Sandra Boatemaa Kushitor 

Academic Editor

PLOS ONE